# Photocatalytic conversion of sugars to 5-hydroxymethylfurfural using aluminium(III) and fulvic acid

Tana Tana[1,2], Pengfei Han[2,3] ✉, Aidan J. Brock[2], Xin Mao[2], Sarina Sarina [2,4], Eric R. Waclawik [2], Aijun Du[2], Steven E. Bottle [2] & Huai-Yong Zhu [2] ✉

5-hydroxymethylfurfural (HMF) is a valuable and essential platform chemical for establishing a sustainable, eco-friendly fine-chemical and pharmaceutical industry based on biomass. The cost-effective production of HMF from abundant C6 sugars requires mild reaction temperatures and efficient catalysts from naturally abundant materials. Herein, we report how fulvic acid forms complexes with $Al^{3+}$ ions that exhibit solar absorption and photocatalytic activity for glucose conversion to HMF in one-pot reaction, in good yield (~60%) and at moderate temperatures (80 °C). When using representative components of fulvic acid, catechol and pyrogallol as ligands, 70 and 67% HMF yields are achieved, respectively, at 70 °C. $Al^{3+}$ ions are not recognised as effective photocatalysts; however, complexing them with fulvic acid components as light antennas can create new functionality. This mechanism offers prospects for new green photocatalytic systems to synthesise a range of substances that have not previously been considered.

The production of chemicals from renewable biomass feedstock has attracted increasing interest over the last three decades, in recognition of the need to transition from an unsustainable economy based on fossil fuel resources to a sustainable, bio-based economy incorporating waste minimisation by design[1–8]. Biomass-derived chemical feedstocks have clear potential to replace fossil-based resources and to decrease environmental impacts in the manufacture of chemical products. For example, the hydrolysis of cellulose, the most abundant component (30–50%) in lignocellulosic biomass, is a rich source of C6 sugars. Processing of these sugars can produce high-value chemicals such as 5-hydroxymethylfurfural (HMF)[5,6,9,10]. Challenges in this reaction process arise in the efficient conversion of sugars to HMF at an industrial scale[1,5,7,9,11–14]. HMF is a precursor to key biochemicals such as 2,5-furandicarboxylic acid, which is considered an essential building block both for the biochemical industry and as a monomer source for biodegradable polymers[1–7,15]. Catalysis is the key to making the HMF synthesis a green and sustainable process. Lewis-acid type metal salts can catalyse the isomerization of aldoses (such as glucose) to ketoses (e.g., fructose), which is the rate-limiting step for the generation of HMF from aldoses[1,2,5–7,11–14,16]. Brønsted acid-catalysed dehydration of ketose sugars to HMF is kinetically more favourable than the transformation of aldose sugars under the same reaction conditions[16]. However, glucose is considered a more suitable feedstock for HMF production due to its high abundance[1,3,6,11–14]. As a representative example, Lewis acid-catalysed isomerization of glucose to fructose, followed by Brønsted acid-catalysed dehydration of fructose to HMF, has been studied extensively[1] and has considerable potential as a green route to this high-value compound[1,3,11,14,16–19].

The essential requirement for the economically viable large-scale conversion of sugars is the use of abundant materials to prepare catalysts that act efficiently at moderate temperatures. Aluminium ions are more abundant, less expensive and less toxic than transition metal

[1]School of Mongolian Medicine, Inner Mongolia Minzu University, Tongliao, Inner Mongolia 028000, China. [2]School of Chemistry and Physics, Queensland University of Technology, Brisbane, QLD 4001, Australia. [3]College of Chemistry and Chemical Engineering, Hunan University, Changsha 410082, China. [4]School of Chemical and Biomolecular Engineering, Faculty of Engineering, The University of Sydney, Camperdown, NSW 2037, Australia. ✉e-mail: pengfeihan@hnu.edu.cn; hy.zhu@qut.edu.au

ion Lewis acid catalysts. As such, AlCl₃ has been extensively studied as a Lewis acid catalyst for transforming sugars[5,9,13,14,16,18,20,21]. With AlCl₃, good HMF yields (52–69%) have been achieved in a variety of reaction systems[14,18,21]. Typically, however, these reactions require elevated temperatures (between 130 °C and 170 °C), and high concentrations of Brønsted acid are necessary for the reaction to proceed. The practical limitation of using higher temperatures and concentrated acids is that HMF is unstable under such conditions, forming unwanted byproducts such as levulinic acid[3,22]. Lower reaction temperatures and lower acidities are also more cost-effective and environmentally benign.

In the present study, we found that mixtures of Al³⁺ and fulvic acid (FA, a class of abundant natural compounds found in soil, peat, and coal) in dimethyl sulfoxide (DMSO) solution, with the proper ratios of FA to Al³⁺ exhibit significant visible light absorption. The absorption promotes catalytic performance for D-glucose dehydration at 80 °C without additional acids. Light-enhanced acid catalysis represents an innovative strategy to develop green processes for acid-catalysed chemical transformations.

## Results

### Producing HMF from glucose using FA-Al³⁺ mixture as the photocatalyst

Ideally, photocatalysis could drive such chemical transformations with light under mild conditions[23–25]. The problem for photocatalysis with simple Al³⁺ species is that their limited light absorption (Fig. 1a) and poor redox chemistry are not conducive to current photocatalytic mechanisms. However, we note that when components of FA chelate with Al³⁺, the resultant complexes can exhibit decent light absorption in the UV-visible region (Fig. 1a).

Increasing the Al³⁺ concentration from 0.01 to 0.02 M does not cause significant change the absorption spectrum. In contrast, doubling the concentration of FA from 2 g L⁻¹ to 4 g L⁻¹ (2FA-Al³⁺ line) increases the absorption over the range of 350–600 nm. The expanded optical absorption range may enhance the catalytic properties of the coordination complexes.

Pilot studies indicate that visible-light irradiation of a solution of aluminium nitrate, FA and glucose in dimethyl sulfoxide (DMSO) solvent significantly increases HMF yield compared with a dark reaction

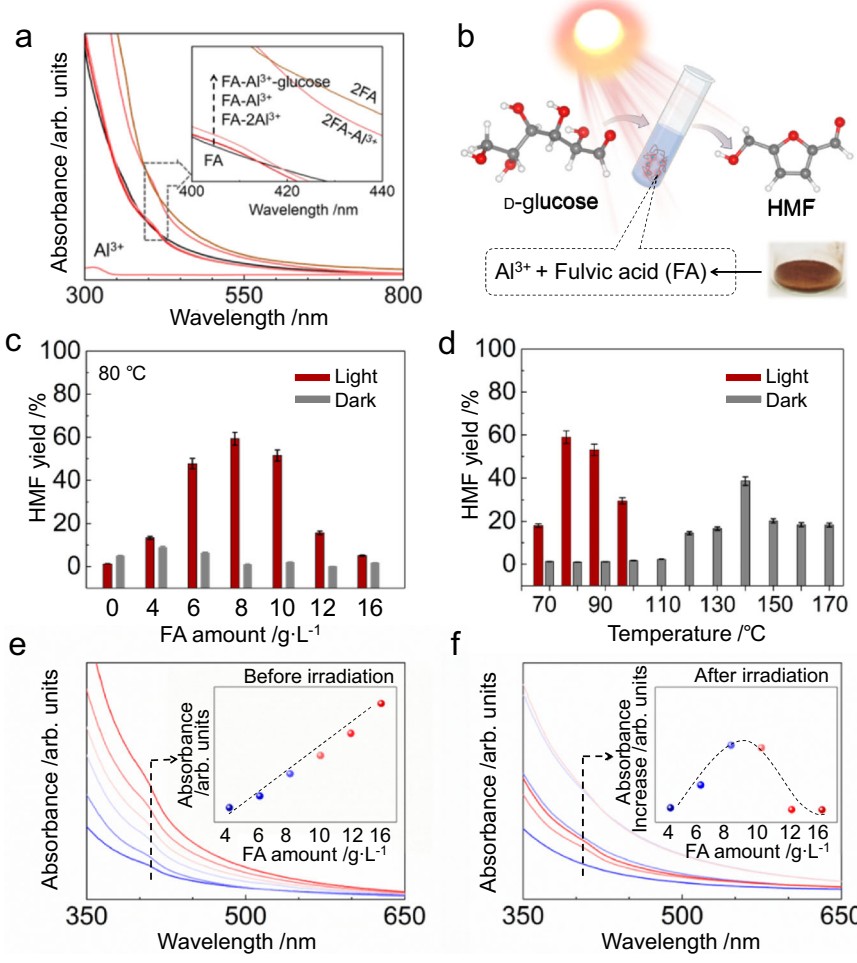

**Fig. 1 | Light absorption of FA, Al³⁺ complexes and catalytic performance of the Al³⁺ complexes systems. a** The UV-vis absorption spectra, from bottom to top, were obtained for the following solutions: 0.01 M Al(NO₃)₃·9H₂O; 2 g L⁻¹ fulvic acid (labelled FA); 2 g L⁻¹ fulvic acid and 0.01 M Al(NO₃)₃·9H₂O (FA-Al³⁺); 2 g L⁻¹ fulvic acid and 0.02 M Al(NO₃)₃·9H₂O (FA-2Al³⁺); 2 g L⁻¹ fulvic acid, 0.01 M Al(NO₃)₃·9H₂O and 1 M glucose (FA-Al³⁺-glucose); 4 g L⁻¹ fulvic acid and 0.01 M Al(NO₃)₃·9H₂O (2FA-Al³⁺), as well as 4 g L⁻¹ fulvic acid (labelled 2FA). DMSO was used as the solvent, the solutions were diluted 3 times with DMSO prior to UV-vis analysis. The insert illustrates the spectral distinctions. **b** Schematic illustration of the photoreaction system. **c** The impact of FA concentration on HMF yield under light irradiation and

in the dark. Reaction conditions: 0.01 M Al(NO₃)₃ • 9H₂O, 4-16 g L⁻¹ FA, 0.1 M D-glucose in 2 mL of DMSO, halogen light intensity of 1.2 W cm⁻² (400–800 nm wavelength), reaction temperature of 80 °C, argon atmosphere at a pressure of 1 bar. **d** The influence of reaction temperature on the dark reaction. The error bars in panels (**c**) and (**d**) associated with HMF yield represent the standard error of three sets of unique measurements. **e, f** UV-vis absorption spectra of the reaction mixture obtained with varying FA concentrations, while maintaining constant Al(NO₃)₃ • 9H₂O (0.01 M) and glucose (0.1 M) concentrations in DMSO before and after 20 h light irradiation. The reaction mixtures were diluted 5-fold with DMSO prior to analysis. The insert in panel (**f**) illustrates the increases in light absorbance.

**Table 1 | Catalytic performance of the photocatalysts for the transformation of sugars to HMF under visible-light irradiation and in the dark[a]**

| Entry | Aluminium salt | Sugar | Fulvic acid (g L$^{-1}$) | Condition | HMF yield (%) |
|---|---|---|---|---|---|
| 1 | Al(NO$_3$)$_3$·9H$_2$O | 1a | 8.0 | Light | 59 |
| 2 | | | | Dark | 6 |
| 3 | Al(NO$_3$)$_3$·9H$_2$O | 1a | / | Light | 1.3 |
| 4 | | | | Dark | 5 |
| 5 | / | 1a | 8.0 | Light | 0.5 |
| 6 | | | | Dark | 0.1 |
| 7 | / | 1a | / | Light | 0 |
| 8 | | | | Dark | 0 |
| 9[b] | Al(NO$_3$)$_3$·9H$_2$O | 1a | 8.0 | Light | <0.1 |
| 10 | Al(NO$_3$)$_3$·9H$_2$O | 1b | 8.0 | Light | 73 |
| 11 | | | | Dark | 52 |
| 12 | Al(NO$_3$)$_3$·9H$_2$O | 1c | 8.0 | Light | 60 |
| 13 | | | | Dark | 23 |
| 14 | AlCl$_3$ | 1a | 8.0 | Light | 50 |
| 15 | Al$_2$(SO$_4$)$_3$ | 1a | 8.0 | Light | 0 |
| 16 | Al(CH$_3$COO)$_3$ | 1a | 8.0 | Light | 0 |
| 17[c] | Al(NO$_3$)$_3$·9H$_2$O | 1a | 8.0 | Light | 21 |
| 18[d] | Al(NO$_3$)$_3$·9H$_2$O | 1a | 8.0 | Sunlight | 43 |

Reaction conditions:

[a]0.01 M of aluminium salt, 8.0 g L$^{-1}$ fulvic acid, 0.1 M sugar/DMSO, halogen light intensity of 1.2 W cm$^{-2}$ (400–800 nm wavelength), 80 °C, 20 h, 1 bar argon atmosphere. Dark reactions were conducted at 80 °C.

[b]No glucose was added into the reaction mixture, yet trace amounts of HMF were yielded from carbohydrates derived from FA.

[c]The reaction was conducted in an air atmosphere of 1 bar.

[d]The reaction was conducted under focused sunlight with an intensity of approximately 0.29 W cm$^{-2}$, at 80 °C for 24 h.

(Table 1, entries 1 and 2), as illustrated in Fig. 1b. The main product was identified by $^1$H NMR spectroscopy (Supplementary Fig. 1) as HMF with a structure consistent with previously reported spectra[19].

Control experiments confirm that light irradiation and the combination of both Al$^{3+}$ and FA are critical for this transformation (Table 1, entries 3–9). The absence of FA in the control experiments conducted under light irradiation result in a mere a 1.3% HMF yield, indicating its indispensability for light absorption in this system. In the absence of Al$^{3+}$ cations, minimal glucose conversion was observed, indicating that Al$^{3+}$ cations are an indispensable component of the active catalytic sites in this system. Negligible HMF was detected without glucose in the reaction mixture, indicating that only trace amounts of FA were converted to HMF yield. Furthermore, no HMF was observed in the absence of both catalyst components.

Given that glucose-to-fructose isomerisation is widely accepted as the initial step in Lewis acid-mediated conversion of glucose to HMF[13,26], additional studies using fructose as substrate instead of glucose were conducted. Remarkably, a high yield of 73% upon irradiation was achieved, which is approximately 40% higher than that in the dark

(Table 1, entries 10 and 11). The use of disaccharide sucrose as the substrate afforded a 60% of HMF yield upon irradiation, which is comparable to the yield obtained from glucose dehydration (Table 1, entries 1, 12 and 13). These results suggest that the photocatalyst system can effectively accelerate Brønsted acid-catalysed dehydration of fructose to HMF, while its contribution to the enhancement of glucose-to-fructose isomerisation, which is the rate-limiting step in the reaction pathway[27], is more significant.

Aluminium sulphate exhibits poor solubility in DMSO, thereby impeding the formation of active photocatalytic species. In contrast, the use of aluminium chloride affords substantial HMF yield (50%), as this salt is more soluble in DMSO than its sulphate counterpart and provides a greater concentration of catalytically useful dissolved Al$^{3+}$. The anionic acetate ligands coordinate firmly with the hard Lewis acid aluminium (III)[28], blocking sugar molecules from accessing Al$^{3+}$ and initiating the catalytic reaction (Table 1, entries 14–16). This highlights the critical necessity for precise regulation of ligand and anion selection in this reaction (detailed discussion can be found in Supplementary Section 1). Additionally, DMSO has been proposed to enhance

fructose dehydration by stabilising crucial intermediates, which may contribute to the remarkable yields observed in our system[29,30].

The conversion rate significantly decreases under ambient atmosphere, with only 21% HMF yield observed (Table 1, entry 17). The possible reason for the loss of activity in this case is the oxidation of the polyphenol ligand (sourced from FA), as discussed in Supplementary Information (Section 2).

FA concentration for glucose transformation was optimised by maintaining the initial aluminium nitrate and glucose concentrations (Fig. 1c). Under light irradiation, a maximum HMF yield of 59% was obtained when the FA concentration is 8.0 g L$^{-1}$, which is approximately ten times higher than that achieved in the dark for the same system. However, exceeding the FA concentration of 8.0 g L$^{-1}$ resulted in a decrease in HMF yield. The conversion of glucose to HMF in this reaction system is a one-pot process without requiring additional Brønsted acid for completion. In contrast, conventional HMF synthesis routes necessitate the addition of Brønsted acid alongside with Lewis acid catalysts to facilitate glucose conversion to fructose and HMF[11]. Cutting out the need for an acidifying step is a significant advantage of the photocatalytic system, especially if it can be scaled up. Our experimental results indicate that the optimal temperature for achieving high photocatalytic conversion yield was 80 °C (Supplementary Table 1), which is considerably higher than yields obtained from non-illuminated thermal reactions heated up to 170 °C (Fig. 1d). The moderate reaction temperature can be readily attained by using widely available solar energy. The thermal reaction behaves similarly to reactions conducted without FA (see Supplementary Section 3)[13,16,18,20,21]. Decreased HMF yields at high temperatures are typically attributed to the formation of humin-like higher molecular weight products resulting from the condensation of HMF and sugars[3].

The conversion was initiated following an extended induction period of 4 h of visible-light irradiation at the optimised reaction temperature (Fig. 2a). Upon comparing the UV-Vis spectra before (Fig. 1e) and after the reaction (Fig. 1f), it is observed that the light absorption of each solution increases significantly during the course of the reaction, except for the solutions with 12 and 16 g L$^{-1}$ of FA (insert in Fig. 1f). Adding glucose in the solution of Al$^{3+}$ and FA in DMSO also increases the absorption of light with wavelengths shorter than 440 nm (Fig. 1a). Therefore, the induction period may be attributed to the slow formation of an active catalytic species under light irradiation, specifically the complexes formed by FA components and glucose coordinated to Al$^{3+}$ ions. For example, hydroxyls in both FA and glucose will release protons when coordinating with Al$^{3+}$. This process occurs during the induction period and the light absorption changes after the reaction (Fig. 1e, f).

The absorbance of 12 and 16 g L$^{-1}$ solutions does not change notably after light irradiation, and the HMF yield at these FA concentrations is low. The concentrations of dominant species [Al(H$_2$O)$_6$]$^{3+}$ and hydrolysed species [Al(OH)(aq)]$^{2+}$ (Supplementary Figs. 3–5 and Section 4) in these solutions are lower than in solutions with lower FA content. This result is of interest as in high FA-content solutions, more coordination between FA components and Al$^{3+}$ ions can impede substrate sugar access to the coordination sphere of Al$^{3+}$, ultimately reducing photocatalytic ability. In other words, excess FA ligands appear to suppress glucose access to the coordination sphere of Al$^{3+}$, thereby inhibiting glucose conversion. Furthermore, the HMF yield increases with illuminance in this system (Fig. 2b), indicating that the light irradiation plays a role in promoting HMF production.

The ideal system would utilise a renewable energy source in conditions that closely mimic natural fluctuations such as intermittent sunlight irradiation. As such, we conducted an experiment using lens-focused sunlight at 80 °C for a total of 24 h (Supplementary Fig. 6 and Section 5), split across four 6-hour periods of clear, focused sunlight, with the illuminance measured at -0.29 W cm$^{-2}$ (2.9 suns). An HMF yield of 43% was obtained under these conditions (Table 1, entry 18).

## Polyphenols as key building blocks for FA-assisted photocatalysis

FA is a complex that contains various functional groups, such as carboxyl, phenol, carbonyl, methoxy and amines[31–33]. To better understand the active ligand components of FA, we tested simpler model ligands containing these functional groups to find more efficient photocatalysts for the conversion (Fig. 3a). Notably, using either catechol or pyrogallol as the sole ligand under LED light irradiation at 440 nm wavelength with an intensity of 0.45 W cm$^{-2}$ afforded excellent HMF yields, 70 and 67%, respectively, at 70 °C for 20 h. The highest yields of HMF were also achieved using polyphenols as sole ligands under a broad-spectrum white light (400–800 nm) of 0.9 W cm$^{-2}$ at 70 °C for 20 h (Fig. 3b). The complexes of Al$^{3+}$ with other functional groups exhibit much lower photocatalytic activities (Fig. 3a). Negligible or moderate yields of HMF were achieved at the same reaction temperature for the reactions conducted under the dark (right panel in Fig. 3a).

The properties of the systems using polyphenols are similar to those using FA. The product yields under light irradiation are about twenty times higher than those observed for the same systems conducted in the dark (Fig. 3b, c). There exist optimal ligand concentrations that maximise HMF yields under light irradiation, while increasing the concentration above the optimal concentration decreases these yields (Fig. 3b). The HMF yield increases with

a

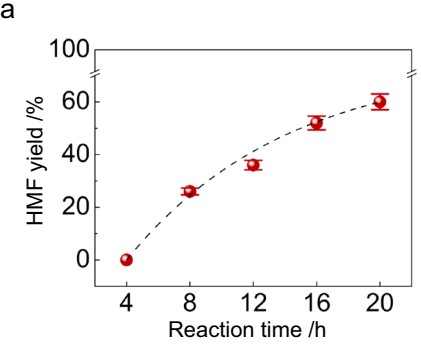

b

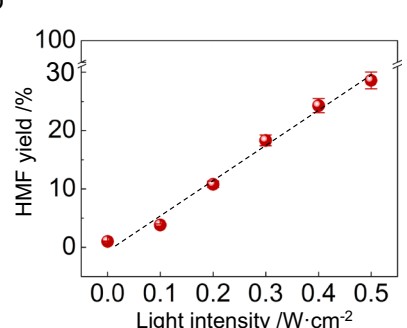

**Fig. 2 | The influence of reaction time and light intensity on the photocatalytic performance of FA-Al$^{3+}$ catalyst for glucose conversion. a** The time course of the photocatalytic reaction. Reaction conditions: 0.01 M Al(NO$_3$)$_3$ • 9H$_2$O, 8 g L$^{-1}$ FA, 0.1 M D-glucose in 2 mL of DMSO, 1.2 W cm$^{-2}$ of halogen light intensity (400–800 nm wavelength), 80 °C reaction temperature, an argon atmosphere of 1 bar. **b** The impact of light intensity on the HMF yield. The reaction conditions in panel a were replicated, with exception of utilising a homemade 440 ± 5 nm LED light as the light source. The error bars associated with HMF yield represent the standard error of three sets of unique measurements.

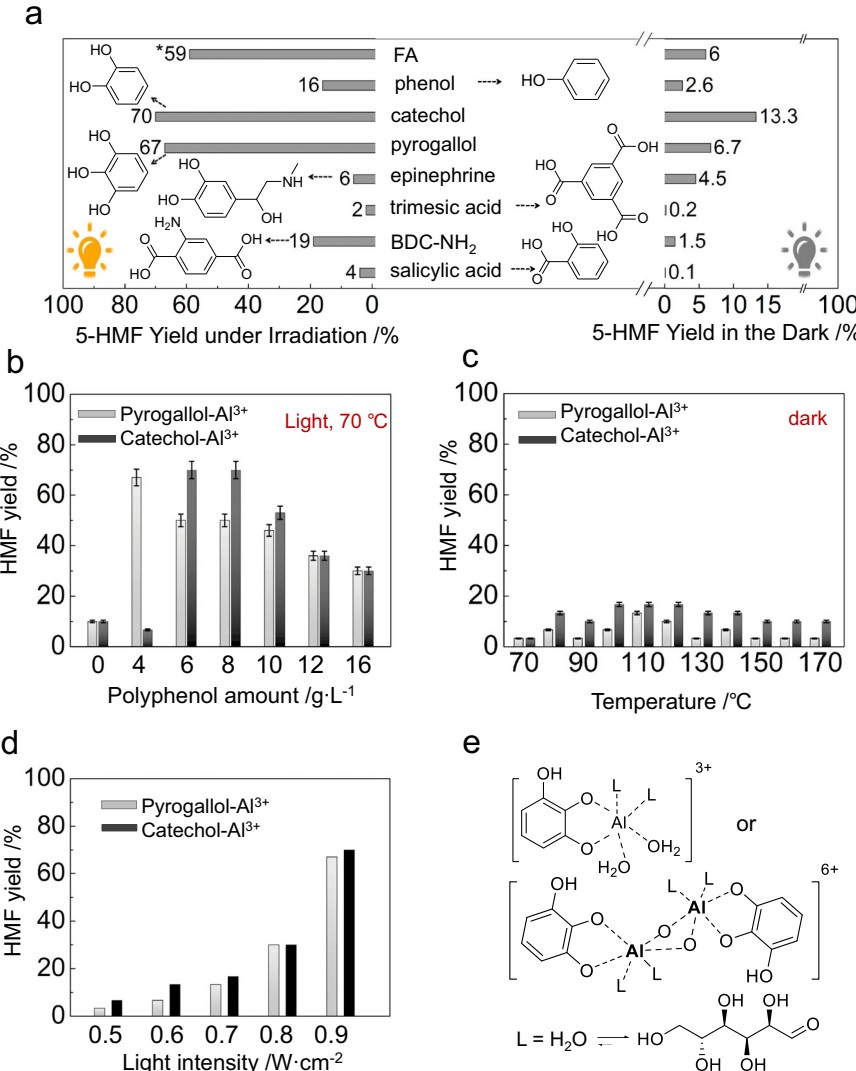

**Fig. 3 | Identification of the key building blocks of FA as the ligands in Al³⁺ complex for photocatalytic conversion of glucose. a** Impact of different additives on the catalytic performance of Al³⁺ salt for the transformation of D-glucose to HMF under visible light irradiation. Reaction conditions: $0.01 M Al(NO_3)_3 \cdot 9H_2O$, 8 mg of additives, 0.1 M D-glucose/DMSO, the reactions proceed under $440 \pm 5$ nm LED light with a light intensity of 0.45 W cm⁻² at 70 °C for 20 h in an argon atmosphere of 1 bar. *1.2 W cm⁻² of light intensity (400–800 nm wavelength), 80 °C. The data on the left and right represent the HMF yields under light irradiation and in the dark, respectively. **b** Impact of polyphenols concentration on HMF yield under light irradiation and in the dark. Reaction conditions: $0.01 M Al(NO_3)_3 \cdot 9H_2O$, 4–16 g L⁻¹ catechol or pyrogallol, 0.1 mmol of D-glucose in 1 mL of DMSO, 0.9 W cm⁻² of light intensity (400–800 nm wavelength), 70 °C reaction temperature, 1 bar of an argon atmosphere. **c** impact of reaction temperature on HMF yield in the dark reaction. The error bars in panels (**c**) and (**d**) associated with HMF yield represent the standard error of three sets of unique measurements. **d** The impact of light intensity on the photocatalytic reaction. The concentrations of polyphenols are 4 and 8 g L⁻¹ for pyrogallol and catechol, respectively. Other conditions were the same as those for panel (**b**) unless specified. **e** The proposed photocatalytic active complex contains a chelating polyphenolate, two ligands (denoted as L) and additional capping ligands, such as water molecules, to complete the coordination sphere. Glucose molecules can replace the ligands (L) by chelating with the Al³⁺ cations.

illuminance in this catalytic system (Fig. 3d). These similarities between the systems corroborate that the polyphenols in FA make the most significant contribution to photocatalysis.

Figure 4 compares the reaction temperature and HMF yield of the present study with that of recently published glucose conversion research. The polyphenol-Al³⁺ catalysts, without additional acids under visible light irradiation, exhibit comparable or even better catalytic performance at a much lower reaction temperature than the thermal catalysts with additional hydrochloric acid.

The complexes of Al³⁺ and polyphenols the best catalyst for converting glucose to HMF as they achieved excellent yields at low reaction temperatures without requiring additional acid. These low temperatures can be readily attained by widely used solar heaters, a mature technology.

We infer that the polyphenols can be converted to polyphenolates, which coordinate with Al³⁺ through a bidentate chelating mode to form a five-membered ring (Fig. 3e). Glucose also coordinates with Al³⁺ through similar bidentate chelating modes[13], allowing it to replace weaker ligands such as DMSO or water and coordinate with Al³⁺ for reaction on a time scale commensurate with the catalytic process. Also, in the systems with high polyphenol or FA concentrations (Figs. 3b and 1c), the high concentration of ligands (other than glucose) can impede glucose coordination and conversion. This indicates a competition between the ligand and glucose for coordination to Al³⁺ ions, which may result in a slow formation of the active catalytic species during the induction period. Following deprotonation, the resulting polyphenolates coordinate with Al³⁺ ions. Visible-light irradiation during the induction period, may also trigger dissociation of

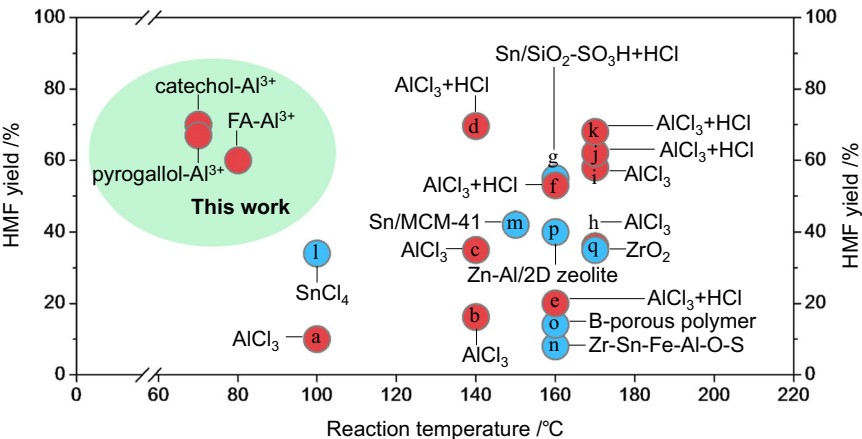

**Fig. 4 | Performance of photocatalysts in this work and thermal catalysts reported in the literature.** HMF yields and reaction temperatures of glucose conversion achieved with FA(polyphenol)-$Al^{3+}$ photocatalysts and thermal catalysts reported in the literature. Blue marks represent the systems using DMSO as the solvent, and red marks indicate the systems using $Al^{3+}$ as the catalysts. **a** ref. 1; **b** ref. 38; **c** ref. 14; **d** ref. 39; **e** ref. 40; **f** ref. 41; **g** ref. 42; **h** ref. 43; **i** ref. 44; **j** ref. 21; **k** ref. 9; **l** ref. 45; **m** ref. 46; **n** ref. 47; **o** ref. 48; **p** ref. 49 and **q** ref. 50.

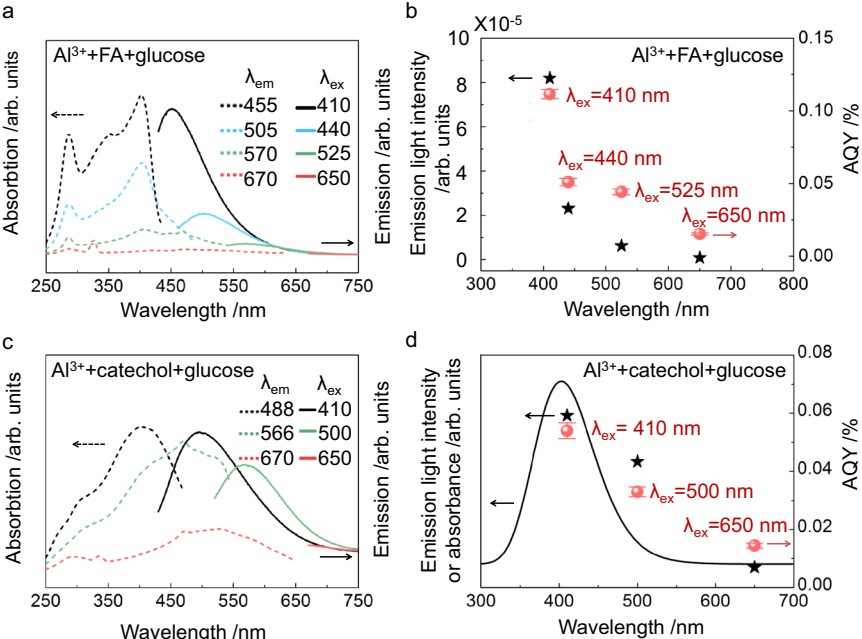

**Fig. 5 | Photoluminescence spectroscopy and wavelength study of FA(catechol)-$Al^{3+}$ catalyst for the transformation of glucose to HMF.** **a**, **c** photoluminescence spectroscopy. The dot-dashed and solid lines are the excitation ($\lambda_{ex}$) and emission ($\lambda_{ex}$) spectra of the complexes, respectively. **b**, **d**, variation of the emission intensity and AQY with wavelength. Reaction conditions: 0.01 M Al(NO$_3$)$_3$ • 9H$_2$O, 8 g L$^{-1}$ FA or catechol, 0.1 M glucose/DMSO, reaction at 80 °C for 20 h in 1 atm argon atmosphere. For the results shown in panel (**a**), the mixture was heated at 50 °C for 30 min and diluted 100 times with DMSO before the measurements. While in panel (**c**), the mixture was irradiated at 0.5 W cm$^{-2}$ of light intensity (400−800 nm) at 70 °C for 20 h and diluted 40 times with DMSO before the measurements. The spectra were measured at room temperature. In panels (**b**) and (**d**), the AQY results were from the reactions conducted at 80 °C for 20 h, and light sources with different wavelengths (410 ± 5 nm, 440 ± 5 nm, 500 ± 5 nm, 525 ± 5 nm, 650 ± 5 nm) were used as light sources, respectively. The light intensities were 0.2 W cm$^{-2}$ and 0.3 W cm$^{-2}$ in panels (**b**) and (**d**), respectively. The peak intensities of the emissions upon excitation with light of wavelengths $\lambda_{ex}$ in panels (**a**) and (**c**) are presented in panels (**b**) and (**d**) to compare the trend of intensity variation with wavelength $\lambda_{max}$ and that of the AQY variation. The curve in panel (**d**) is the UV-vis absorption electronic spectrum of the Al-catechol-glucose complex simulated by the TD-DFT procedure. The error bars in panels (**b**) and (**d**) associated with AQY represent the standard error of three sets of unique measurements.

weak coordinating ligands, similar to that observed from transition metal complex catalysts[34], allowing for coordination of glucose. This results in the absorption increase and induction period observed in Fig. 1. The isomerization product fructose leaves the complexes as it rapidly dehydrates to yield HMF.

Analysis of the photoluminescence emission spectra of the catalytic systems supports a mechanism in which glucose transformation occurs at the light-excited state of the $Al^{3+}$-complexes, as evidenced by examining emission following excitation at wavelengths used to transform sugar into HMF (Fig. 5a).

Detailed photoluminescence emission spectra of the catalytic system and its separate components are discussed in Supplementary Information (Supplementary Fig. 7 and Section 6). Figure 5b shows that the maximum emission intensity varies with the excitation wavelength

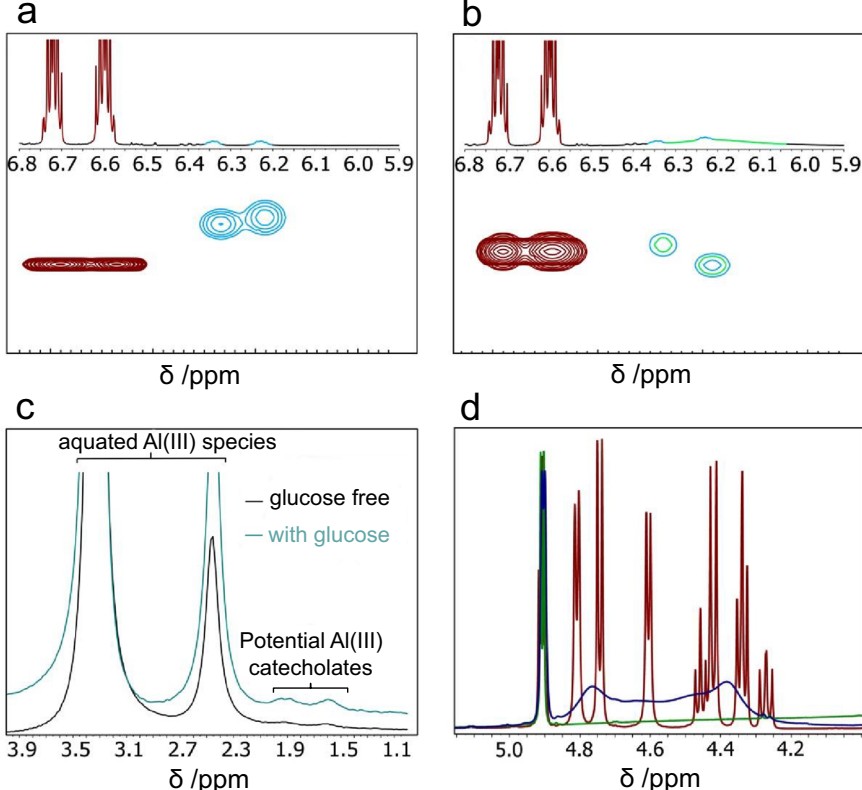

**Fig. 6 | NMR investigations of aluminium nitrate/catechol solutions. a** $^1$H NMR and DOSY spectrum of aluminium nitrate nonahydrate (0.01 M) and catechol (0.02 M) in DMSO-d$_6$. **b** $^1$H NMR and DOSY spectrum of aluminium nitrate nonahydrate (0.01 M), catechol (0.02 M) and D-glucose in DMSO-d$_6$. A colour key is provided for the proposed assignments (red: catechol, blue: aluminium catecholate, green: glucose). **c** Corresponding $^{27}$Al NMR spectra of the solutions used in panels (**a**) and (**b**). **d** $^1$H NMR spectra of (red) D-glucose (0.1 M); (blue) D-glucose (0.1 M) and aluminium nitrate nonahydrate (0.01 M); (green) D-glucose (0.1 M), aluminium nitrate nonahydrate (0.01 M) and catechol (0.02 M) in DMSO-D$_6$. Glucose -OH region is shown only.

$\lambda_{ex}$ (Fig. 5a) in the same trend of the HMF yield varying with the irradiation wavelength. The observed AQY is the highest upon irradiation at the shoulder peak at 410 nm, which is six times higher than under 650 nm irradiation. Significant activity is still observed at wavelengths above 500 nm. FA is a complex mixture; longer wavelength irradiation will likely excite different FA components that absorb across the entire spectrum, so this broad activity is not unexpected. The high activity at wavelengths around 410 nm indicates the presence of catalytically active ligand species comprised of smaller conjugated compounds.

The complexes of Al$^{3+}$-polyphenol-glucose absorb long wavelengths (650 nm or longer), but the HMF yield is low under the longer wavelength excitation (Fig. 5b and 5d). The yield is highly dependent on the wavelength. The results reveal a minimum energy threshold for excited electronic states of the Al$^{3+}$ complexes: only those generated by absorption of shorter wavelengths (<650 nm) have sufficient energy for effective glucose conversion.

The molar absorption coefficients ($\varepsilon$) for aluminium complexes with catechol and pyrogallol, calculated using the total Al$^{3+}$ concentration, are 1780 and 6320 L mol$^{-1}$ cm$^{-1}$, respectively, at 400 nm (Supplementary Fig. 8 and Section 7). The large $\varepsilon$ values correlate well with the case of the light-induced ligand-to-metal charge transfer (LMCT) in the Al$^{3+}$-complexes they form[35,36]. This phenomenon can cause an electron density change in molecules chelating with Al$^{3+}$ ions[36]. The shifts of electron density from glucose ligand to Al$^{3+}$ ion is expected to promote the Lewis acid-catalysed isomerization of glucose to fructose.

NMR analysis was conducted to obtain information on the active catalytic structure. Al$^{3+}$ cations in the solution of Al(NO$_3$)$_3$ in pure DMSO are octahedrally coordinated (Supplementary Figs. 9, 10, and

Section 8) and correlate with the aquated species [Al(OH)(H$_2$O)$_5$]$^{2+}$ and [Al(H$_2$O)$_6$]$^{3+}$. Upon the addition of catechol, two small peaks adjacent to the initial signals appear in the $^{27}$Al NMR spectrum, which may correspond to the octahedral aluminium catecholate species. These peaks do not appreciably change with the addition of glucose, indicating only transient complexation between glucose and Al$^{3+}$ cations (Fig. 6c). A small symmetrical catechol signal (0.02 H relative to free catechol) is observed upfield of the parent catechol peaks in $^1$H NMR (Fig. 6a). DOSY NMR (Fig. 5a) shows a decrease in diffusion coefficient from $4.14 \times 10^{-10}$ to $2.78 \times 10^{-10}$ m$^2$ s$^{-1}$, which is consistent with the formation of a coordination complex. The addition of glucose (Fig. 5b) does not appear to cause a major difference in the diffusion behaviour of this species; however, accurate determination of the diffusion coefficient is challenging due to signal overlap with glucose $^1$H signals, which reduces the observed value. The addition of glucose (1 or 10 equivalents) to the Al$^{3+}$-catechol solution appears to cause significant broadening or loss of the glucose hydroxyl proton signals compared to glucose or glucose-Al$^{3+}$ solutions (Fig. 6d), indicating some form of non-specific interaction between the Al$^{3+}$ complexes and glucose. DOSY NMR shows that catechol and glucose signals did not diffuse at the same rate, suggesting no persistent formation of Al$^{3+}$-catechol-glucose complexes. Therefore, it is likely that transient Al$^{3+}$-catechol-glucose species are formed during the catalytic process under light irradiation, with the remainder of the aluminium (III) coordination sites occupied by solvent ligands under non-reacting conditions. Pyrogallate ligand is expected to form similar structures.

Additional NMR studies on the interaction of FA with Al$^{3+}$ are discussed in Supplementary Information (Supplementary Fig. 11 and Section 8). The formation of complexes in Al$^{3+}$-FA and Al$^{3+}$-FA-glucose

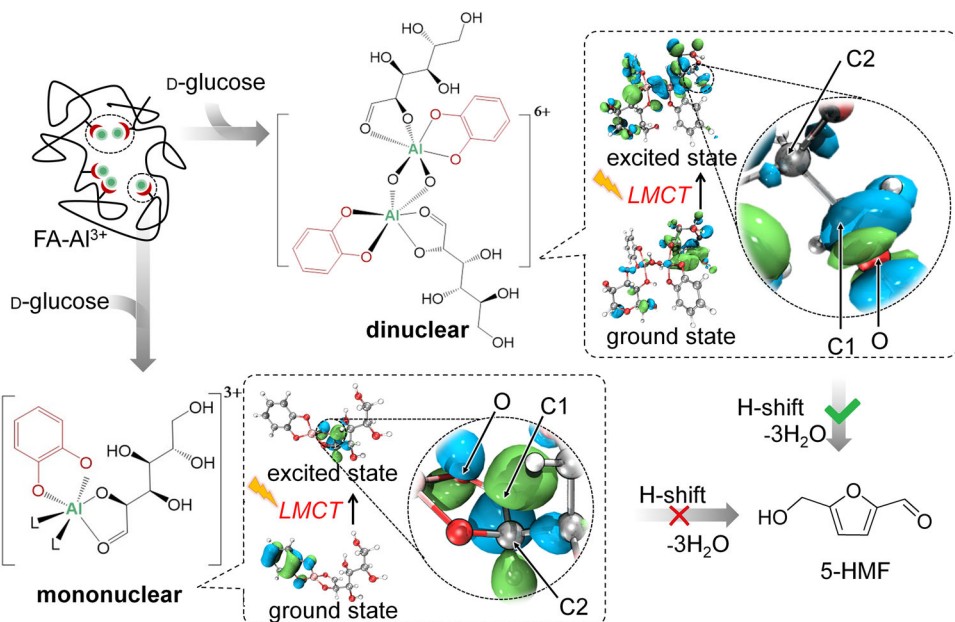

**Fig. 7 | Proposed tentative reaction mechanism for glucose conversion with aluminium (III)-catechol-glucose complexes.** Coordinated ligand L is bidentate bridging D-glucose, weakly coordinated water, DMSO solvent or non-catalytic, weakly bound FA components. The ground and excited states for aluminium (III) complexes were computed at B3LYP/6-31 + G(d) level. The green and blue regions represent positive and negative phases of the molecular orbital surface, respectively.

mixtures is supported by Small-angle X-ray Scattering (SAXS) analysis (Supplementary Fig. 5), which explains the change in light absorption during the reaction.

The coordination sphere of the $Al^{3+}$ cation contains at least a chelating polyphenolate and additional capping ligands to complete its coordination. We propose that in the polyphenolate-$Al^{3+}$ system, these capping ligands include water or hydroxide as well as weakly coordinating species (solvent or nitrate anions), which can be readily displaced by glucose during the reaction, resulting in complexes of the forms shown in the scheme in Fig. 3e. In this system, water serves as a capping agent and provide Brønsted-acidic $H^+$ ions. Polyphenols also release protons, forming polyphenolate to coordinate to $Al^{3+}$ ions.

The set of symmetric catechol $^1H$ resonances observed in Fig. 6 supports the formation of symmetric complexes with the structure shown in Fig. 3e. The presence of only a small quantity of aluminium catecholate under ambient conditions suggests that the liability of both substrate and product is critical to the function of this catalytic system, as it allows for efficient turnover and minimises catalyst deactivation.

### The dinuclear Al(III) complex structure has superior activity over the mononuclear structure

With the acquired structural knowledge of the coordination complexes, we performed a computational study using the aluminium (III)-catechol complexes in Fig. 3e as models to comprehend the distribution of electron density in their excited electronic states and understand the mechanism of action. As shown in Fig. 7, the ground electronic state of the monometallic complex is mostly centred on the catechol ligand and $Al^{3+}$ cation, with electron density transferring mostly to the glucose ligand and $Al^{3+}$ cation in the excited electronic state. However, this behaviour does not correlate well with the photocatalytic performance of the reaction. In contrast, the bimetallic hydroxide-bridged complex appears to match well with the expected photocatalytic mechanism. In the excited state, the electron density of a coordinated glucose molecule shifts to a catechol ligand and one $Al^{3+}$ cation, resulting in the LMCT effect. The change in electron density facilitates the H-shift in glucose molecules and thus glucose isomerization, providing a pathway for effective sunlight photocatalysis.

A similar phenomenon of structure-based electron density shifting can be observed when pyrogallol is replaced as the ligand (Supplementary Fig. 12 and Section 9). This behaviour corresponds well with the observed reactivity and the wavelength dependence of the reaction, where various polyphenol analogues will absorb at different wavelengths.

Based on experimental and computational data, we propose a general mechanism for the photocatalytic enhancement of this reaction (Fig. 7), in which light excitation promotes the necessary 1,2-hydride shift or subsequent C2 proton abstraction for aldose-to-ketose tautomerization. In the non-photocatalytic variant of this reaction, it has been suggested that coordination of aluminium(III) with the carbonyl oxygen polarises the C = O bond, enhancing the tautomerization[13,37]. In our proposal, photoexcitation further polarises the coordinated glucose C = O bond relative to the non-excited species. The shift of electron density to C1 is more favourable in the binuclear structure under an excited state, facilitating proton transfer to the C1 position (i.e., the tautomerization), similar to what was previously proposed for a $Cr^{3+}$-based system[19].

In summary, it has been demonstrated abundant and inexpensive $Al^{3+}$ ions and fulvic acid component ligands can combine to form novel and efficient photocatalysts for sugar conversion under mild reaction conditions without additional Brønsted acid. By combining naturally sourced light-absorbing components with $Al^{3+}$ ions, new opportunities are opened up for converting sugars at an industrial scale via a green process and developing new photocatalysts. The coordination of light antennas to metal ions to form active photocatalytic complexes is expected to stimulate the future design of inexpensive and efficient photocatalysts for synthesising a range of substances that have not previously been considered.

## Methods
### Chemicals
The chemicals were purchased from commercial suppliers and used as provided. The supplier and purity of the chemicals are indicated in the

brackets. D-fructose (Sigma-Aldrich, >99%), D-glucose (AnalaR, >99%), sucrose (Sigma-Aldrich, >99.5%), 5-hydroxymethyl furfural (Sigma-Aldrich, >99%), methanol (Fisher, HPLC assay), dimethyl sulfoxide (DMSO, Ajax Finechem, >99%), aluminium (III) nitrate nonahydrate (Ajax Finechem, >98%), aluminium chloride (Greagent, ≥99%), aluminium sulfate octadecahydrate (Aladdin, 99.95%), aluminium acetate (Macklin, ≥98%), fulvic acid soluble powder (FA, Agtech Natural Resources, >92%), phenol (Adamas, 99%), catechol (Adamas, 99%), pyrogallol (Adamas, 99%), epinephrine (Adamas, 97%), trimesic acid (Adamas, >99%), 2-aminoterephthalic Acid (Adamas, >98%), salicylic acid (Adamas, 99%), DMSO-δ (Sigma-Aldrich), Ar (Supagas, >99.99%).

#### Photocatalytic reactions and products analysis
The photocatalytic reaction was conducted in a light reaction chamber. A 10 mL Pyrex glass tube was used as the reaction container. After adding the reactants and catalyst, the tube was filled with argon and sealed with a rubber septum cap. Then the tube was placed on a magnetic stirrer, illuminated under a halogen lamp (Philips Industries: 500 W, wavelength in the range 400–800 nm), and stirred at a controlled reaction temperature. The reaction temperature in the dark was maintained the same as that for the reaction under irradiation using an oil bath placed on a magnetic stirrer. The tube was wrapped with aluminium foil to avoid exposure to the reaction to light.

The specimens of the reaction mixture were taken at designed reaction periods, diluted with water by 50 times and filtered through a Millipore filter (pore size 0.45 μm). The target product, HMF, was quantified with commercial HMF as an external standard using high-performance liquid chromatography. The HMF product was analyzed by HPLC (Agilent 1100 equipped with a UV detector). The mobile phase (0.5 mL min⁻¹) was methanol/water (90:10 v/v). HMF was detected by the UV detector (280 nm) with a Waters C8 column at room temperature. The HMF yield is defined as follows:

$$Y_{HMF} = (\text{moles of HMF obtained}/\text{moles of starting glucose or fructose}) \times 100\%.$$

$$Y_{HMF} = (\text{moles of HMF obtained}/\text{moles of starting sucrose})/2 \times 100\%.$$

To calculate carbon balance, residual D-glucose and the intermediates in reaction mixture after 20 h light irradiation and in the dark were analysed using HPLC (Agilent 1260 equipped with an Evaporative Light-scattering Detector).

#### Catalyst characterisation
**UV-Visible analysis.** Cary 5000 UV-Vis-NIR spectrometer (Agilent) was used to collect the diffuse reflectance UV-visible (DR-UV-vis) spectra of the solid samples. Cary 60 spectrometer (Varian) was used to collect absorption spectra for liquid samples. The emission and excitation spectra were recorded on a Quanta Master 4 fluorescence spectrometer using quartz cuvettes loaded with 2 mL of sample

The molecular weight distribution of fulvic acid in DMSO was analyzed with Gel permeation chromatography (Agilent PL-GPC50, Agilent) using a PLgel Olexis column (300 mm × 7.5 mm, Agilent). The mobile phase was DMSO at a flow rate of 1 mL/min, 100 μL injection volume, and molecular weights were determined by comparison with Pullulan standards.

**NMR analysis.** ¹H and ²⁷Al NMR spectra for liquid samples were recorded using a 600 MHz NMR Bruker Advance spectrometer (Bruker, U.S.) at room temperature. The spectral widths were 5 kHz and 20 kHz for the ¹H and ¹³C dimensions, respectively. The number of the collected complex points was 1024 for the ¹H dimension with a recycle delay of 1.5 s. ²⁷Al NMR spectral width was 40 kHz, and the number of the collected complex points was 4096. 2D DOSY NMR spectra were collected using default DOSY pulse sequences on a 400 MHz Bruker

Advance spectrometer (Bruker, U.S.). Sixteen spectra were collected across a linear distribution of gradient strength from 2 to 98%.

The SAXS experiments were performed at the SAXS/WAXS beamline of the Australian Synchrotron (ANSTO), Melbourne, Australia. The samples were investigated using the small-/wide-angle X-ray scattering beamline (900 mm camera length using Pilatus 1M and 200 K detectors, transmission mode), and Scatterbrain software was used for the analysis. The X-ray had a beam with a wavelength of 1.07812 Å (11.5 keV). The distance from the sample to detector was calibrated with a silver behenate standard.

The electronic structures and UV-vis absorption electronic spectrum of Al³⁺ complexes were theoretically simulated by the Time Dependent Density Functional Theory (TD-DFT) procedure. The geometry of the complex and TD-DFT calculation was optimised at B3LYP/6-31 + G(d) level and performed using Gaussian 16, Revision C.01, software package.

## Data availability
The authors declare that the data generated in this study are provided in the paper and its supplementary information. Other raw and relevant data during the study are available from the corresponding authors upon request.

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

## Acknowledgements

We acknowledge financial support from the Australian Research Council. (DP190100499, DP200102652, DP210103357 and DE190101450). PH acknowledges the National Natural Science Foundation of China (No. 22002038) and the Natural Science Foundation of Hunan Province (No. 2023JJ40120). Prof. Shuaijun Pan from Hunan University is acknowledged for SAXS experiments.

## Author contributions

T.T. and P.H. found the photocatalytic activity of Al3+ complexes and performed the experiments. H.Y.Z., P.H. and A.B. proposed the mechanism of the photocatalytic system. A.B. conducted the NMR

study. X.M. and A.D. conducted the DFT simulation study. H.Y.Z., T.T., P.H., A.B., E.W., S.E.B. and S.S. wrote the manuscript.

## Competing interests

The authors declare no competing interests.
