## [Peer Review File · Nature Communications]

Photocatalytic conversion of sugars to 5-hydroxymethylfurfural using aluminium(III) and fulvic acidReviewers' Comments:

Reviewer #1:

Remarks to the Author:

Dear Editors, Dear Authors,

Professors Han, Zhu and colleagues reported an interesting photoconversion of sugars into 5-hydroxymethylfurfural (HMF) promoted by aluminium(III) salts in the presence of fulvic acid (FA) or phenolic-based ligands.

By adding Al^{3+} cation and fulvic acid to glucose the authors observed the formation of species capable to absorb visible light. These species promote, under irradiation, the formation of HMF from glucose, fructose, and sucrose. The use of Lewis and Bronsted acids for the transformation of sugars in HMF is a well-established procedure but usually required high temperature. The authors discover that by irradiation at 70°C of the reaction mixture composed by $AlCl_3$, FA and the substrate in DMSO, HMF was obtained with comparable or better yields compare to the reaction conducted in the dark at higher temperature. From these results a deep study of the mechanism of the reaction was conducted with different techniques. The presence of phenolic moieties in the FA suggested the formation of phenoxy-aluminium complexes able to promote the transformation by coordination of the glucose. By the use of simple model phenolic ligands for Al^{3+} highest yields in the formation of HMF were achieved avoiding the use of the FA. Computational and NMR studies on these simplest complexes suggested the formation of a dinuclear $Al(III)$ complex that in its excited state promote the reaction.

The present manuscript presents noteworthy results that go beyond the established literature and the data obtained will be important to the field in a way that will move the field forward. The possibility to convert sugars into HMF with high yields in straightforward manner by the use of simple reagents open the way to new processes for biomass conversion in large scale.

The approach to the problem by the authors is excellent as the quality of the data reported to support the discussion and conclusions (both in the main text and in the supporting information).

Although the high degree of complexity of the system, the analytic approach to the understanding the process is complete and include different advanced techniques.

The manuscript needs to be revised in its structure to give to the readers a better comprehension of the results obtained (see comments below).

The procedures reported in the main text and in the supporting information give enough detail of the work to be reproduced.

Previous literature is appropriately inserted in the manuscript and full analysis of the published literature was conducted by the authors for comparison.

Best Regards

- Table 1 should be inserted and discussed at the beginning after the discussion of Figure 1a/b. The influence of reaction time and light intensity and the other sugar studied have to follow the table 1.
- Insert the right enantiomer of sugars with stereochemical indicators.
- Is the reaction mixture homogeneous?
- UV-Vis spectrum of a solution of 4 g L⁻¹ fulvic acid should be collected and inserted in Figure 1a for comparison with the spectrum of 2FA- Al^{3+} .
- Table S2 should be inserted in the main text.
- Results obtained in Table 1 entries 3 and 4 should be discussed in the text to highlight the role of FA.
- In figure 3a the pH of the different phenols tested is reported. Is it the pH of a water solution of the phenol? Is it the pH of the reaction mixture? If you reported these data, have to be discussed in the text.
- Figure 3a: in my opinion the graph should be putted by side with the corresponding graph under irradiation. This will be highlighting the effect of the temperature on the reaction outcome.
- Please report specification for the spectrofluorometer used for emission and excitation spectra.
- Figure S4: report the ²⁷Al NMR analysis of reaction mixtures after 20h without irradiation.

- Section 6 supporting information: The authors claim that system is affected by oxygen due to the oxidation of catechol ligand to semiquinone. The only experimental evidence describe is a change in the color of the complex. Is it possible to have an instrumental analysis to support that conclusion? For example, NMR or mass spectroscopy.
- Supporting information lines 230-239: this phrase should be relative to section 8 not section 9.

Reviewer #2:

Remarks to the Author:

The conversion of glucose to HMF remains one of the interesting topic for the development of sustainable chemistry. In the present work, the authors reported one new reaction system, that is, Aluminum species with light, for the conversion of glucose to HMF at low temperature of 80°C, and an acceptable yield of about 60% was obtained. Many characterization methods were used to characterize the reaction system and correlations were proposed. My concerns are the followings:

1. Concerning the use of aluminum salt, the authors are suggested to use directly and uniformly the chemical formula of the substance used throughout the manuscript (including in Figures, Tables, and supporting materials) according to their experimental fact, for example, $\text{Al}(\text{NO}_3)_3$, or $\text{Al}(\text{NO}_3)_3 \cdot 9\text{H}_2\text{O}$, or $\text{Al}(\text{NO}_3)_3 \cdot 6\text{H}_2\text{O}$, since it might cause different states of the aluminum species in the system. When discussing the states of aluminum species, the authors are also suggested to consider the possible hydrolysis or decomposition of $\text{Al}(\text{NO}_3)_3 \cdot 9\text{H}_2\text{O}$ under the reaction conditions, if the authors used $\text{Al}(\text{NO}_3)_3 \cdot 9\text{H}_2\text{O}$ as the aluminum source, the hydrolysis and/or decomposition might occur readily in the heating processes. Furthermore, the formation of HMF is unavoidably accompanied by stoichiometrical water formation, which might affect the existence of aluminum species. These facts should be considered in the related discussion.

2. For solvent effects, DMSO could also interact with proton, forming more active catalytic species to enhance the dehydration of fructose to HMF (ACS Catal. 2017, 7, 2199–2212), which could not be eliminated under the present conditions. The authors are suggested to discuss this issue by referring the above mentioned literature in addition to [9] mentioned in the supporting information.

3. The calculation methods for the yield of HMF from different substrates needs to be provided, especially that from sucrose. Furthermore, the authors reported only the yield of HMF, how about the conversion of substrates and carbon balance of the reactions? These should also be reported.

4. Page 4, lines 81-82, please check if the sentence, "Conversion of glucose to fructose by Brønsted acid addition is a key intermediate step in the conventional HMF synthesis route.11", is proper here, since the conversion of glucose to fructose conversion here is said to be catalyzed by Lewis acids.

5. Page 6, lines 116-119, the sentence, "This result can be explained if the Brønsted acid content of the reaction mixture (sourced from the FA) is considered sufficient to hydrolyse the glycosidic bond of the sucrose dimer into its glucose and fructose constituents for further conversion to HMF.", needs to be reconsidered, since the Brønsted acid in this system could also be originated form the hydrolysis of $\text{Al}(\text{NO}_3)_3 \cdot 9\text{H}_2\text{O}$.

6. In Table 1, entry 4, the HMF yield obtained without light irradiation (5%) is much higher than that with light, why? The authors are suggested to give some explanation. It is not clear if the reaction carried out without light needs also induction time, why?

7. Concerning the reaction mechanism, it could be seen in Table S2 that when fructose was used as the starting material with light, the HMF yield was obviously lower than when glucose was used, while the use of sucrose gave much higher HMF yield. Thus, it seems that there existed reaction route other than via fructose route from glucose to HMF, otherwise the direct use of fructose should give higher yield if the formation of fructose were necessary for glucose conversion. The authors are suggested to consider these facts when elaborating the mechanistic aspects.

8. Proofreading needs to be checked. Typical examples are:

(1) Page 3, line 60, please check if it is better change "the ratios of FA to Al^{3+} in certain a range" to "the proper ratios of FA to Al^{3+} ".

(2) Page 7, line 130, it is better change "with wavelengths short than 440 nm" to "with wavelengths shorter than 440 nm".

(3) Page 15, lines 289-291, please check if the sentence, "The intensity of the observed absorbance using pyrogallol or catechol ligands (Supplementary Fig. 7 and Section 8) correlates well with the ease of the light-induced ligand-to-metal charge transfer (LMCT) in the aluminium complexes they form.", needs to be improved.

(4) Page 20, line 380, the statement, "working under near-natural ambient conditions", is not proper, since the presence of air decreased the efficiency.

Reviewer #3:

Remarks to the Author:

The paper presents a novel topic, the photocatalytic conversion of glucose to HMF in the presence of Al³⁺ complexes of phenol derivatives. The photocatalytic effect improves the conversion respect to the standard activity of the Al salt, allowing to work at lower temperature with higher selectivity and yields. The paper has a not clear aspect: the role of DMSO in the proposed mechanism. The authors report: "DMSO has also been proposed to enhance the dehydration of fructose by stabilizing key intermediates, which may contribute to the high yields observed in our system", but in the proposed mechanism and in the discussion DMSO role is never clarified. This solvent is not a green medium, thus it would be necessary to report few runs with the same system in water. The activity of in water would be much more interesting, Al salts being active in the conversion of glucose in water to HMF or, at least, in DMSO/water. Moreover, the results obtained with the other sugars (fructose and sucrose) deserve to be inserted in the text and not in the Supplementary Material, because they help to understand the double effect of isomerization and dehydration. It is not clear the result obtained with sucrose, similar to that reached with fructose. It is a not common result because generally the performances with sucrose are intermediate, have the authors well verified these data? The paper needs the required revisions/improvements.

Point-to-point responses to the comments of the reviewers

Reviewer #1 (Remarks to the Author):

Comments:

Table 1 should be inserted and discussed at the beginning after the discussion of Figure 1a/b. The influence of reaction time and light intensity and the other sugar studied have to follow the table 1.

Response:

Thank the reviewer for the comment. We moved Table 1 and the discussion after Figure 1 a and b. following the suggestion.

Comment:

Insert the right enantiomer of sugars with stereochemical indicators.

Response:

We appreciate the advice and corrected the enantiomer of sugars in Table 1, Fig. 7, and Fig. S11 in the revised version.

Comment:

Is the reaction mixture homogeneous?

Response:

Yes, FA-Al³⁺ catalysts are soluble in DMSO.

Comment:

UV-Vis spectrum of a solution of 4 g L⁻¹ fulvic acid should be collected and inserted in Figure 1a for comparison with the spectrum of 2FA-Al³⁺.

Response:

Thank the reviewer, and the comment has been taken. We collected the UV-Vis spectrum of 4 g L⁻¹ fulvic acid in DMSO (2FA) and inserted it in Figure 1a as below.

Comments:

Table S2 should be inserted in the main text.

Response:

We appreciated the reviewer's comment and incorporated the data from Table S2 in the previous version into Table 1 of the revised version.

Comments:

Results obtained in Table 1 entries 3 and 4 should be discussed in the text to highlight the role of FA.

Response:

Thank the reviewer and the suggestion is taken. We have discussed and highlight the role of FA in the main text as follows:

“Control experiments confirm that light irradiation and the combination of both Al^{3+} and FA are critical for this transformation (Table 1, entries 3-9). The absence of FA in the control experiments conducted under light irradiation result in a mere a 1.3% HMF yield, indicating its indispensability for light absorption in this system.”

Comments:

In figure 3a the pH of the different phenols tested is reported. Is it the pH of a water solution of the phenol? Is it the pH of the reaction mixture? If you reported these data, have to be discussed in the text.

Response:

The pH values were determined by measuring the reaction mixture, which was diluted with 20-fold of deionized water before the measurement. Although we attempted to monitor the influence of pH, the most active catalytic systems had a narrow range of the pH values between 3.1 and 3.6. We removed the data from the revised manuscript as they was not discussed.

Comments:

Figure 3a: in my opinion the graph should be putted by side with the corresponding graph under irradiation. This will be highlighting the effect of the temperature on the reaction outcome.

Response:

Thank the reviewer for the valuable advice. To better demonstrate the effect of the light irradiation, we replaced the pH values of the reaction mixture with HMF yields in the dark. The revised Fig. 3a below shows significant increases in the HMF yield under light irradiation.

Fig. 3 | Identification of the key building blocks of FA as the ligands in Al³⁺ complex for photocatalytic conversion of glucose. a, Impact of different additives on the catalytic performance of Al³⁺ salt for the transformation of d-glucose to HMF under visible light irradiation. Reaction conditions: 0.01 M Al(NO₃)₃·9H₂O, 8 mg of additives, 0.1 M d-glucose/DMSO, the reactions proceed under 440±5 nm LED light with a light intensity of 0.45 W cm⁻² at 70 °C for 20 h in an argon atmosphere of 1 bar. *1.2 W cm⁻² of light intensity (400-800 nm wavelength), 80 °C. The data on the left and right represent the HMF yields under light irradiation and in the dark, respectively. b, Impact of polyphenols concentration on HMF yield

under light irradiation and in the dark. Reaction conditions: 0.01 M $\text{Al}(\text{NO}_3)_3 \cdot 9\text{H}_2\text{O}$, 4-16 g L^{-1} catechol or pyrogallol, 0.1 mmol of d-glucose in 1 mL of DMSO, 0.9 W cm^{-2} of light intensity (400-800 nm wavelength), 70 °C reaction temperature, 1 bar of an argon atmosphere. c, impact of reaction temperature on HMF yield in the dark reaction. d, The impact of light intensity on the photocatalytic reaction. The concentrations of polyphenols are 4 and 8 g L^{-1} for pyrogallol and catechol, respectively. Other conditions were the same as those for panel (b) unless specified. e, The proposed photocatalytic active complex contains a chelating polyphenolate, two ligands (denoted as L) and additional capping ligands, such as water molecules, to complete the coordination sphere. Glucose molecules can replace the ligands (L) by chelating with the Al^{3+} cations.

Comments:

Please report specification for the spectrofluorometer used for emission and excitation spectra.

Response:

We apologize for the missing information. We have included it in the revised manuscript:

“The spectra were recorded on a Quanta Master 4 fluorescence spectrometer using quartz cuvettes loaded with 2 mL of sample”.

Comments:

Figure S4: report the ^{27}Al NMR analysis of reaction mixtures after 20h without irradiation.

Response:

Thank the reviewer for the comment. We have conducted the ^{27}Al NMR analysis on reaction mixtures obtained from glucose conversion to HMF using Al^{3+} and different FA concentrations after 20 h without light irradiation, as shown below. The new data has been added to Fig. S4 for comparison.

Comments:

Section 6 supporting information: The authors claim that system is affected by oxygen due to the oxidation of catechol ligand to semiquinone. The only experimental evidence describe is a change in the color of the complex. Is it possible to have an instrumental analysis to support that conclusion? For example, NMR or mass spectroscopy.

Response:

Thank the reviewer for the comment. We conducted LC-MS analysis on the reaction mixture (see the figure below) and found that semiquinone was produced due to the oxidation of catechol ligand during the reaction conducted in an air atmosphere. The figure has been included in Section 2 in the revised version.

Figure S2. Detection of the product obtained in the reaction under aerobic conditions.refers to Section 6 by LC-MS (Agilent 1290 uplc- Agilent qt of 6550). Reaction conditions: 0.01 M of aluminium salt, 8.0 g L⁻¹ catechol, 0.1 M glucose/DMSO, 1.2 W cm⁻² of light intensity (400-800 nm wavelength), 80 °C, 20 h, 1 atm air atmosphere. m/z (ESI) calculated for (C₆H₄O₂)[M+H]⁺:109.02, found: 109.02.

Comments:

Supporting information lines 230-239: this phrase should be relative to section 8 not section 9.

Response:

We apologise for the mistakes. They have been corrected.

Reviewer #2 (Remarks to the Author):

The conversion of glucose to HMF remains one of the interesting topic for the development of sustainable chemistry. In the present work, the authors reported one new reaction system, that is, Aluminum species with light, for the conversion of glucose to HMF at low temperature of 80°C, and an acceptable yield of about 60% was obtained. Many characterization methods were used to characterize the reaction system and correlations were proposed. My concerns are the followings:

Comments:

1. Concerning the use of aluminum salt, the authors are suggested to use directly and uniformly the chemical formula of the substance used throughout the manuscript (including in Figures, Tables, and supporting materials) according to their experimental fact, for example, $\text{Al}(\text{NO}_3)_3$, or $\text{Al}(\text{NO}_3)_3 \cdot 9\text{H}_2\text{O}$, or $\text{Al}(\text{NO}_3)_3 \cdot 6\text{H}_2\text{O}$, since it might cause different states of the aluminum species in the system. When discussing the states of aluminum species, the authors are also suggested to consider the possible hydrolysis or decomposition of $\text{Al}(\text{NO}_3)_3 \cdot 9\text{H}_2\text{O}$ under the reaction conditions, if the authors used $\text{Al}(\text{NO}_3)_3 \cdot 9\text{H}_2\text{O}$ as the aluminum source, the hydrolysis and/or decomposition might occur readily in the heating processes. Furthermore, the formation of HMF is unavoidably accompanied by stoichiometrical water formation, which might affect the existence of aluminum species. These facts should be considered in the related discussion.

Response:

We appreciate the insightful comment. It is true that the states of the aluminum species in the system may vary with aluminium salts, which partially explains why that different yields were observed under identical conditions. As shown in Table 1, that the best HMF yield is achieved by using $\text{Al}(\text{NO}_3)_3 \cdot 9\text{H}_2\text{O}$. The most active catalytic systems were found to work best within a narrow pH range of 3.1 to 3.6 after dilution, with the actual pH being ≤ 3 . At this level, aluminum cations undergo moderate hydrolysis and can form dinuclear Al(III) complex but little large oligomers (Clays and Clay Miner. 41, 598-607, 1993). We also investigated the impact of water on the photocatlytic performance of the FA- Al^{3+} catalyst (see the figure below). Our finding indicate that adding a small amount of water to the reaction mixture can optimise HMF yield. However, higher level of water in the system may reduce HMF yield due to the lower stability of HMF in aqueous environment (Catal. Commun. 51 (2014) 5–9.).

Figure S14b. The impact of water content on HMF yield under light irradiation. Reaction conditions: 0.01 M Al(NO₃)₃·9H₂O, 8 g L⁻¹ FA, 0.1M d-glucose in 2 mL of DMSO or H₂O-DMSO cosolvent, 1.2 W cm⁻² of halogen light intensity (400-800 nm wavelength), 80 °C reaction temperature, 1 bar of argon atmosphere.

Comments:

2. For solvent effects, DMSO could also interact with proton, forming more active catalytic species to enhance the dehydration of fructose to HMF (ACS Catal. 2017, 7, 2199–2212), which could not be eliminated under the present conditions. The authors are suggested to discuss this issue by referring the above mentioned literature in addition to [9] mentioned in the supporting information.

Response:

Thank the reviewer for the comment on a crucial issue. We have discussed the solvent effects by referencing relevant literature (ACS Catal. 2017, 7, 2199–2212) as suggested., This source has been cited in our supporting information, section 11.

Comments:

3. The calculation methods for the yield of HMF from different substrates needs to be provided, especially that from sucrose. Furthermore, the authors reported only the yield of HMF, how about the conversion of substrates and carbon balance of the reactions? These should also be reported.

Response:

Thanks for the reminder. The target product, 5-HMF, was quantified using commercial 5-HMF as an external standard and analysed by high-performance liquid chromatography (HPLC) with a UV detector. We have been included the calculation methods for the yield of HMF from different substrates in the manuscript as suggested. To fructose and glucose conversion

$$\text{yield} = (\text{concentration of HMF after reaction}/\text{concentration of sugars before reaction}) \times 100\%$$

while to sucrose conversion

$$\text{yield} = (\text{concentration of HMF after reaction}/\text{concentration of sugars before reaction}) \times 50\%.$$

To calculate carbon balance, residual D-glucose and the intermediates in reaction mixture after 20 h light irradiation and in the dark were analysed using HPLC (Agilent 1260 equipped with an Evaporative Light-scattering Detector). The results are presented in the table below and also included in SI. Based on conversion of substrates and products yield, the carbon balance for the light reaction is calculated to be 81.5%, while that for dark reaction is 86%. This suggests that there are some by-products, such as humin formation during the D-glucose conversion.

Table 1. Mass balance of the transformation of D-glucose to HMF under visible-light irradiation and in the dark

Entry	Condition	Conversion(%)	Fructose(%)	HMF(%)	Carbon balance (%)
1	Light	83.5	0	65	81.5
2	Dark	38	13	11	86

Reaction conditions: a 0.01 M of $\text{Al}(\text{NO}_3)_3 \cdot 9\text{H}_2\text{O}$, 8.0 g L^{-1} fulvic acid, 0.1 M D-glucose/DMSO, 1.2 W cm^{-2} of light intensity (400-800 nm wavelength), 80 °C, 20 h, 1 atm argon atmosphere. Dark reactions were conducted at 80 °C.

Comments:

4. Page 4, lines 81-82, please check if the sentence, “Conversion of glucose to fructose by Brønsted acid addition is a key intermediate step in the conventional HMF synthesis route.¹¹”, is proper here, since the conversion of glucose to fructose conversion here is said to be catalyzed by Lewis acids.

Response:

Thanks for the question. We apologise for the confusion caused by our previous statement and have revised it below:

“The conversion of glucose to HMF in this reaction system is a one-pot process without requiring additional Brønsted acid for completion. In contrast, conventional HMF synthesis routes necessitate the addition of Brønsted acid alongside with Lewis acid catalysts to facilitate glucose conversion to fructose and HMF¹¹. Cutting out the need for an acidifying step is a significant advantage of the photocatalytic system, especially if it can be scaled up.”

Comments:

5. Page 6, lines 116-119, the sentence, “This result can be explained if the Brønsted acid content of the reaction mixture (sourced from the FA) is considered sufficient to hydrolyse the glycosidic bond of the sucrose dimer into its glucose and fructose constituents for further conversion to HMF.”, needs to be reconsidered, since the Brønsted acid in this system could also be originated from the hydrolysis of $\text{Al}(\text{NO}_3)_3 \cdot 9\text{H}_2\text{O}$.

Response:

Thank the reviewer for the comment, we removed “sourced from the FA” from this sentence.

Comments:

6. In Table 1, entry 4, the HMF yield obtained without light irradiation (5%) is much higher than that with light, why? The authors are suggested to give some explanation. It is not clear if the reaction carried out without light needs also induction time, why?

Response:

There is no FA ligand in the systems, so light irradiation cannot promote the reaction utilising the LMCT effect as discussed in the manuscript. The reaction of these systems proceeds via a conventional thermal process, and light irradiation weakens its performance due to an unclear effect that possibly reduces the interaction between DMSO and glucose. The induction time can be attributed to the low reaction temperature of 80 °C.

Comments:

7. Concerning the reaction mechanism, it could be seen in Table S2 that when fructose was used as the starting material with light, the HMF yield was obviously lower than when glucose was used, while the use of sucrose gave much higher HMF yield. Thus, it seems that there existed reaction route other than via fructose route from glucose to HMF, otherwise the direct use of fructose should give higher yield if the formation of fructose were necessary for glucose conversion. The authors are suggested to consider these facts when elaborating the mechanistic aspects.

Response:

Thanks for bringing up the question. We used different reaction conditions to convert fructose and glucose. For the fructose reaction, we used 0.001 M of $\text{Al}(\text{NO}_3)_3 \cdot 9\text{H}_2\text{O}$ and 0.8 mg/mL of FA as the catalyst. To avoid any potential confusion, we conducted the fructose conversion under the identical conditions to those employed for glucose conversion (0.01 M of $\text{Al}(\text{NO}_3)_3 \cdot 9\text{H}_2\text{O}$ and 8 mg/mL of FA). As expected, using fructose results in a significantly higher yield of HMF compared to glucose (refer to updated data in Table 1).

Comments:

8. Proofreading needs to be checked. Typical examples are: (1) Page 3, line 60, please check if it is better change “the ratios of FA to Al^{3+} in certain a range” to “the proper ratios of FA to Al^{3+} ”.

Response:

Thank the reviewer and the advice is taken. We carefully revised the manuscript and made corrections.

Comments (2):

Page 7, line 130, it is better change “with wavelengths short than 440 nm” to “with wavelengths shorter than 440 nm”.

Response:

We appreciate the reviewer’s advice and have made the correction based on to the comment.

Comments (3)

Page 15, lines 289-291, please check if the sentence, “The intensity of the observed absorbance using pyrogallol or catechol ligands (Supplementary Fig. 7 and Section 8) correlates well with the

ease of the light-induced ligand-to-metal charge transfer (LMCT) in the aluminium complexes they form.”, needs to be improved.

Response:

This comment has been acknowledged. The sentence has been improved as follows:

“The molar absorption coefficients (ϵ) for catechol and pyrogallol, calculated using the total Al^{3+} concentration, are 1780 and 6320 $L\ mol^{-1}\ cm^{-1}$, respectively, at 400 nm (Supplementary Fig. S8 and Section 7). The large ϵ values correlate well with the ease of the light-induced ligand-to-metal charge transfer (LMCT) in the Al^{3+} -complexes they form^{49,50}. This phenomenon can cause an electron density change in molecules chelating with Al^{3+} ions⁵⁰”

Comments (4):

Page 20, line 380, the statement, “working under near-natural ambient conditions”, is not proper, since the presence of air decreased the efficiency.

Response:

We appreciate the reviewer’s comment and have revised the sentence as “it has been demonstrated abundant and inexpensive Al^{3+} ions and fulvic acid component ligands can combine to form novel and efficient photocatalysts for sugar conversion under mild reaction conditions without additional Brønsted acid.”

Reviewer #3 (Remarks to the Author):

The paper presents a novel topic, the photocatalytic conversion of glucose to HMF in the presence of Al^{3+} complexes of phenol derivatives. The photocatalytic effect improves the conversion respect to the standard activity of the Al salt, allowing to work at lower temperature with higher selectivity and yields. The paper has a not clear aspect: the role of DMSO in the proposed mechanism. The authors report: "DMSO has also been proposed to enhance the dehydration of fructose by stabilizing key intermediates, which may contribute to the high yields observed in our system", but in the proposed mechanism and in the discussion DMSO role is never clarified. This solvent is not a green medium, thus it would be necessary to report few runs with the same system in water. The activity of in water would be much more interesting, Al salts being active in the conversion of glucose in water to HMF or, at least, in DMSO/water. Moreover, the results obtained with the other sugars (fructose and sucrose) deserve to be inserted in the text and not in the Supplementary Material, because they help to understand the double effect of isomerization and dehydration. It is not clear the result obtained with sucrose, similar to that reached with fructose. It is a not common result because generally the performances with sucrose are intermediate, have the authors well verified these data? The paper needs the required revisions/improvements.

Response:

Thank the reviewer for the comment. It has been reported in literature that DMSO plays a role in elevating the energy of HMF’s lowest unoccupied molecular orbital, resulting in the formation of HMF as the primary product (ChemSusChem 2014, 7, 117 – 126.). In our study we did not discover any new function of DMSO and briefly cited relevant literature. While acknowledging

that “this solvent is not a green medium”, we attempted to replace it with other solvents, but have yet to make meaningful progress. Therefore, the focus of this manuscript is on the photocatalysts of Al^{3+} ions and fulvic acid of its components.

We conducted the glucose conversion in the cosolvent system. As shown in the figure below, the addition of other organic solvents such as isopropanol, ethanol, and 1-butanol significantly decreased HMF yield, with almost no HMF obtained after the addition of large amount water. We will endeavor to find an environmentally friendly solvent to replace DMSO as the solvent.

Figure S11a. The result of glucose conversion in the cosolvent system. Reaction conditions: 0.01 M of aluminium salt, 8.0 g L⁻¹ fulvic acid, 0.1 M d-glucose, 1 mL of DMSO and 1 mL of the other medium (water, isopropanol, ethanol or 1-butanol) were used as the cosolvent, 1.2 W cm⁻² of light intensity (400-800 nm wavelength), 80 °C, 20 h, 1 atm argon atmosphere.

As the reviewer suggested, we included the results of the reaction with fructose and sucrose in the manuscript. In previous version, we used different reaction conditions to convert fructose and glucose. For the fructose reaction, we used 0.001 M of $\text{Al}(\text{NO}_3)_3 \cdot 9\text{H}_2\text{O}$ and 0.8 mg/mL of FA as the catalyst. We conducted the fructose conversion under identical conditions to those employed for glucose conversion (0.01 M of $\text{Al}(\text{NO}_3)_3 \cdot 9\text{H}_2\text{O}$ and 8 mg/mL of FA). As expected, using fructose results in a significantly higher yield of HMF compared to glucose (refer to updated data in Table 1).

Reviewers' Comments:

Reviewer #1:

Remarks to the Author:

Dear Editor, Dear Authors,

I have looked at the revised manuscript "Photocatalytic conversion of sugars to 5-hydroxymethylfurfural using aluminium(III) and fulvic acid" that recently I acted as a referee.

The authors answered to all questions that the referees expressed, and the manuscript was modified as consequence.

Some minor points:

- Figure 3a: insert in the axis label "light" and "dark" to clearly indicate the conditions for the results reported in the left and in the right part of the figure.
- Page 16 "The molar absorption coefficients (ϵ) for catechol and pyrogallol...": it should be "The molar absorption coefficients (ϵ) for aluminium complexes with catechol and pyrogallol...".

Reviewer #2:

Remarks to the Author:

The authors have addressed the reviewers' comments carefully. I have no other concerns, and recommend acceptance of it.

Reviewer #3:

Remarks to the Author:

The authors have improved the paper as required following the suggestions of the referees. It can be accepted for publication.

Reviewer #1 (Remarks to the Author):

Dear Editor, Dear Authors,

I have looked at the revised manuscript "Photocatalytic conversion of sugars to 5-hydroxymethylfurfural using aluminium(III) and fulvic acid" that recently I acted as a referee.

The authors answered to all questions that the referees expressed, and the manuscript was modified as consequence.

Some minor points:

- Figure 3a: insert in the axis label "light" and "dark" to clearly indicate the conditions for the results reported in the left and in the right part of the figure.

Response:

Thank the reviewer for the comment. We revised Figure 3a following the suggestion.

- Page 16 "The molar absorption coefficients (ϵ) for catechol and pyrogallol...": it should be "The molar absorption coefficients (ϵ) for aluminium complexes with catechol and pyrogallol...".

Response:

We appreciate the reviewer's advice and have made the correction based on to the comment.

Reviewer #2 (Remarks to the Author):

The authors have addressed the reviewers' comments carefully. I have no other concerns, and recommend acceptance of it.

Reviewer #3 (Remarks to the Author):

The authors have improved the paper as required following the suggestions of the referees. It can be accepted for publication.